# Hierarchical Learning for Modular Robots

**Risto Kojcev, Nora Etxezarreta, Alejandro Hernández and Víctor Mayoral**
Erle Robotics
Venta de la Estrella Kalea, 6
01006 Vitoria-Gasteiz, Araba, Spain
{risto, nora, alex, victor}@erlerobotics.com

## Abstract

We argue that hierarchical methods can become the key for modular robots achieving re-configurability. We present a hierarchical approach for modular robots that allows a robot to simultaneously learn multiple tasks. Our evaluation results present an environment composed of two different modular robot configurations, namely 3 degrees-of-freedom (DoF) and 4DoF with two corresponding targets. During the training, we switch between configurations and targets aiming to evaluate the possibility of training a neural network that is able to select appropriate motor primitives and robot configuration to achieve the target. The trained neural network is then transferred and executed on a real robot with 3DoF and 4DoF configurations. We demonstrate how this technique generalizes to robots with different configurations and tasks.

## 1 Introduction

When performing a complex action, humans do not think or act in the level of granular primitive actions at the individual muscle or joint level. Instead, humans decompose complicated actions in a set of simpler actions. By combining simpler actions or motor primitives, humans can learn more complicated and unseen challenges in a fast and easy way. Moreover, human cognition separates a task at several levels of temporal abstraction. In robotics, the same occurs as complicated tasks are composed of sub-tasks at different levels of granularity ranging from motor primitives to higher level tasks, such as grasping, where different time scales interact. The majority of deep reinforcement learning (DRL) techniques focus on individual actions at single time steps resulting in low sample efficiency when training robots, lack of adaptability to unseen new tasks and low transfer capabilities between related tasks. Hierarchical reinforcement learning methods allow the robot to learn to perform a certain task in the level of macro-actions that are a set of individual actions reducing the search space. This makes the learning process faster and more scalable, and allows the robot to generalize across unseen tasks or environments. Modular robots present a novel approach for building robots where each component of the system is independent but works in symbiosis with the other components, forming a flexible system which can be reconfigured and easily assembled. Compared to traditional robotics as described in Mayoral et al. (2017), the modular robots can facilitate the integration time, ease of re-purposing, and accelerate the development time of different behaviours.

This work focuses on exploring hierarchical techniques for DRL methods, with the goal of allowing training different behaviours on a re-configurable modular robot. The approach presented in this paper evaluates the output of a hierarchical neural network trained for two different configurations of the Scara modular robot, namely 3DoF and 4DoF configuration. Moreover we evaluate the error when each of the configurations of the modular robot tries to achieve different targets.

## 2   PREVIOUS WORK:

In order to develop robots that learn in an efficient and structured manner, temporally-extended actions and temporal abstraction are required. The first hierarchical approach for RL was introduced by Dayan & Hinton (1993). The authors propose a method that speeds up learning by enabling it to happen at multiple resolutions in space and time by introducing management hierarchy. In Dayan & Hinton (1993) work, the high level managers learn how to set tasks to their sub-managers and, in turn, the sub-managers learn how to complete these tasks. The Options framework introduced by Sutton et al. (1999) sets the path towards more structured approaches for reinforcement learning. Options consist of courses of actions extended over different time-scales. In the past, several researchers learned such policies for action by explicitly defining sub-goals and engineered rewards. However, using explicitly defined sub-goals subsequently learned by policies is not scalable when learning complex behaviours. Thus, recent research has focused on automatically learning them, such as: Strategic Attentive Writer for Learning Macro-Actions Vezhnevets et al. (2016), Stochastic Neural Networks for Hierarchical Reinforcement Learning Florensa et al. (2017), Probabilistic inference for determining options in reinforcement learning Daniel et al. (2016) and The option-critic architecture Bacon et al. (2016).

The recent work of Frans et al. (2017) presented a method for learning hierarchies in which they improve the sample efficiency on unseen tasks trough the use of shared policies that are executed for large number of timesteps. The goal of this work is to evaluate the meta-learning shared hierarchies (MLSH) method and its applicability to modular robots. In section 3, we present the theoretical foundation of the MLSH method, and in section 4, we present our experimental evaluation of MLSH for modular robots.

## 3   META-LEARNING SHARED HIERARCHIES (MLSH) FOR MODULAR ROBOTS

The goal of MLSH is to maximize the accumulated reward across a distribution of tasks defined as $P_M$ with a common state and action space

$$maximize_\phi E_{M \sim P_M}[r_0 + r_1 + ... + r_{T-1}] \tag{1}$$

by following a stochastic policy $\pi_{\phi,\theta}(a|s)$, being $\theta$ the task-specific parameters whereas $\phi$ is the shared parameter vector between tasks. In order to find the optimal parameters of the stochastic policy, MLSH finds meaningful sub-policies parametrized by $\phi_k$ that are selected by a master policy parametrized by $\theta$. Given that meaningful sub-policies are discovered, the agent learns to realize new tasks quickly by simply adapting the master policy to the new task.

Learning both policy and sub-policy parameters $\theta$ and $\phi_k$ is divided into two stages, namely the warm-up period and the joint update period. The warm-up period corresponds to the master policy parameters' update period where the sub-policy parameters are fixed and the agent interacts with the environment following the selected sub-policy by the master policy. In the joint update period, both the master policy $\theta$ and the sub-policy selected $\phi_k$ are updated. For more details, the pseudo-code of MLSH can be found in 5.1

Despite the high precision of state-of-the-art DRL methods such as Proximal Policy Optimization (PPO) Schulman et al. (2017), Actor Critic using Kronecker-Factored Trust (ACKTR) Wu et al. (2017), or Trust Region Policy Optimization (TRPO) Schulman et al. (2015), these techniques were developed with a single task and environment in mind. This leads to poor generalization of tasks the robot can perform. We hypothesize that different robots with a similar action and state space share motor primitives that we could leverage when training the robot. Hence, our experiments aim to validate if MLSH manages to converge to the corresponding target position by training the master and sub-policy neural networks on different robot configurations and different target positions.

## 4 EXPERIMENTS

In our experiments, we trained a master policy and corresponding sub-policies that generalize across different robot configurations and target positions by switching the robot configuration and corresponding target position. The simulation environment is described in more details in Appendix 5.2. Our experimental setup consisted of a modular 3DoF robot to be extended by 1 DoF, both in simulation and in the real robot, and two target positions, namely the center of H with point at $[0.3305805, -0.1326121, 0.3746]$ for the 3DoF, and $[0.3305805, -0.1326121, 0.4868]$ for the 4DoF robot. The center of O is set to $[0.3325683, 0.0657366, 0.3746]$ for the 3DoF and to $[0.3325683, 0.0657366, 0.4868]$ for the 4DoF robot. Since our experiment consisted of two different robot configurations and two different target positions, we trained the MLSH network with a number of sub-policies equal to 4, macro duration equal to 5, warm-up time equal to 20 and training time equal to 200. After training the network in simulation, we evaluated the learned MLSH network on the modular robot where the different target positions were reached for the different robot configurations, see Table 1 for details.

| | Target | Euclidean Distance (mm) | |
| --- | --- | --- | --- |
| | | real robot | simulation |
| **3DoF** | Center of O | **31.06±0.15** | $33.69\pm(1.9 \times 10^{-7})$ |
| | Center of H | 60.36±0.12 | 60.07±0.02 |
| **4DoF** | Center of O | 37.02±0.12 | 58.39±0.01 |
| | Center of H | 48.82±0.15 | $46.83 \pm(3.49 \times 10^{-15})$ |

Table 1: Summarized results when executing a network trained with different goals and DoF. The first target is set to the middle of the H, the second target is set to the middle of the O for the 3DoF and 4DoF robots. The trained network is executed both in simulated environment and on the real robot. The MLSH network outputs continuously trajectory points even after convergence, therefore the standard deviation (STD) of the last 10 end-effector points is calculated.

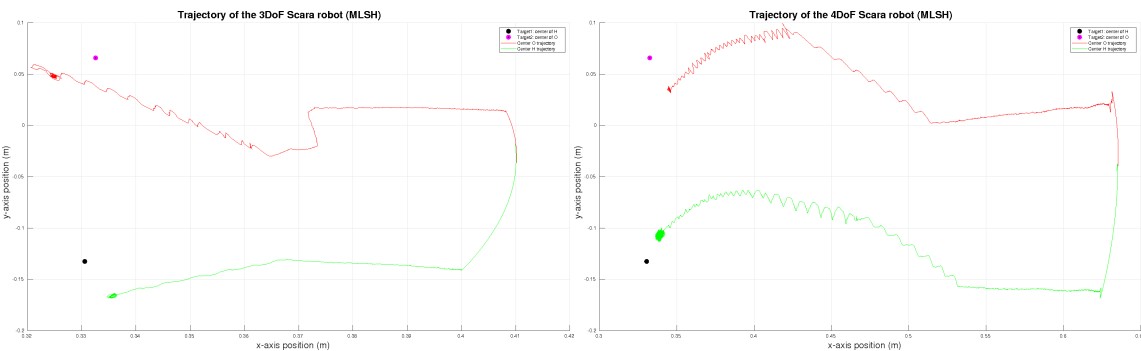

Figure 1: Output of the trajectories for the 3DoF (left) and 4DoF (right) Scara Robot, when loading a previously trained network for different targets.

During the evaluation of MLSH we noticed that, while executing trained network, the master policy selects the same sequence of sub-policies for a particular robot configuration and target position. This behaviour validates the initial claims in the original work of Frans et al. (2017), that the sub-policies are underlying motor primitives that generalize across different tasks. Future work involves investigation of which motor primitives are being learned during training.

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

# 5 APPENDIX

## 5.1 MLSH PSEUDO-CODE

---
**Algorithm 1** MLSH
---
Initialize $\phi$
**repeat**
  Initialize $\theta$
  **for** w=0,1,...W (warmup period) **do**
    Collect D timesteps of experience using $\pi_{\phi,\theta}$
    Update $\theta$ to maximize expected return from $1/N$ timescale viewpoint
  **end for**
  **for** u=0,1,...U (joint update period) **do**
    Collect D timesteps of experience using $\pi_{\phi,\theta}$
    Update $\theta$ to maximize expected return from $1/N$ timescale viewpoint
    Update $\phi$ to maximize expected return from full timescale viewpoint
  **end for**
**until** convergence

---

## 5.2 SIMULATION ENVIRONMENT

As previously presented in Zamora et al. (2016), our novel technique for transferring any network trained in simulation using MLSH techniques to the real robot relies on our extension of the OpenAI gym which is tailored for robotics. For our experiments, we train two modular robots, namely the SCARA 3DoF and 4DoF robots, where the Gazebo simulator and corresponding ROS packages convert the actions generated from each algorithm to appropriate trajectories the robot can execute.

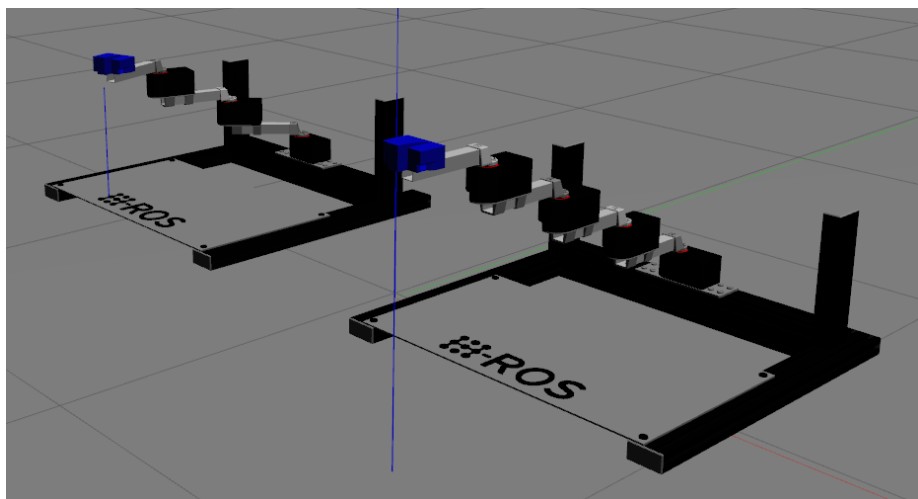

Figure 2: All the training for the 3DoF (illustrated on the left) and 4DoF (illustrated on the right) Scara robots is performed in simulation in our dual environment. Then, the trained network is transferred to the real robot for both configurations and corresponding targets.

