# OpenReview forum: "Hierarchical Learning for Modular Robots"
_ICLR.cc/2018/Workshop — Reject_

### Official Review · AnonReviewer2 · 2018-02-28
**A useful application attempt, but insufficiently described and inconclusive results**

**Rating:** 4
**Confidence:** 4

**Review:**


In this paper, the authors applied the hierarchical "meta-learning shared hierarchies" (MLSH) framework of Frans et al. (2017) to a robotics experiments where two different configurations of the robot (3dof and 4dof) are used to solve the same goal.

The work is clearly and nicely positioned, the attempt is useful, but the description of the results is way too short and no clear conclusion can be drawn from it. Hence I'm inclined towards rejection of this work.

In more details:

- Figure 1 needs to be magnified A LOT before the labels and legend get readable, and a lot of information is missing: where are the starting point and end point? How much error with respect to the target do we get in the end? Is this error acceptable? What do the red and green curve stand for? why are they more different in the 3dof case than in the 4d case? What about the small oscillations?

- The results would be much more informative if they were compared to using any (strong) competing method.

- The second paragraph of Section 2 should make explicit that MLSH is the work of Frans et al. (2017) and not the author's.

typos, minor issues:

- In the "Previous work" section, the authors should use the past tense more consistently.

- in the end of the first paragraph, when titles are used, they should be put into "".

- trough => through

- section 3, 4 => Section


- In Section 3, the criticism of state-of-the-art DRL was already put forward in the introduction. The corresponding references should be moved to the introduction, although they are not mandatory in such a very short paper. You should focus more on what you did and what you obtained.

- with a number of sub-policies equal to 4 => with 4 sub-policies

- The MLSH network outputs continuously trajectory => continuously outputs

---

### Official Review · AnonReviewer3 · 2018-03-08
**validation is potentially interesting;  difficult to evaluate current results**

**Rating:** 3
**Confidence:** 4

**Review:**

The paper presents an experiment that aims to validate the
Meta-Learning Shared Hierarchies (MLSH) for modular robots,
both in simulation and on a robot, for 3 and 4-DOF scara robots.

The abstract was difficult to understand,
as it references simultaneous solving multiple tasks, but then
immediately refers to two types of modular robot. Bringing this sentence forward from the body of
the paper would help: "We hypothesize that different robots with a similar action and state space
share motor primitives that we could leverage when training the robot."

It was unclear to me what the target H and target O tasks were, what the states and actions were,
and how to interpret the results of Table 1 as validating or invalidating the hypothesis.
Of course it is difficult to describe everything in three pages, but I was hoping to see a bit
more in the supplementary pages to help me understand the problem specification.

Is Figure 1 for the robot or the simulation? How does the simulation differ?
Was the policy trained on the simulation or the robot?
What is the significance of the particular trajectory that is shown?
What is the timescale for the results shown in Figure 1?
What alternative methods could be considered for comparison?
How does the method compare to a traditional Scara robot controllers?

Pros and cons:
+ potentially interesting problem, i.e., validating the MLSH ideas
- this reviewer had significant problems understanding the problem being solved,
  the structure of the learned hierarchy, and how to draw conclusions from the
  provided table and figure

---

### Official Review · AnonReviewer1 · 2018-03-08
**The contribution of this paper is not clear.**

**Rating:** 2
**Confidence:** 4

**Review:**


This paper describes how an existing architecture for hierarchical reinforcement learning was applied to simulated robots and then transferred to physical robots.  The hierarchical architecture used is the Meta-Learning Shared Hierarchies method.  The simulated and physical robots were 3-DOF and 4-DOF Scara modular robots.  The task was to move to two different target positions for each of the two robot configurations.

This paper is not written clearly.  I was left uncertain about the relevance of the chosen experiment and most of the experiment's details.  Although this setting of controlling multiple robots moving to multiple targets might potentially benefit from the chosen method, there was not a clear hypothesis being examined and there was no comparison to a baseline system.  The paper provides some of the algorithm's hyperparameters, but it does not provide the experiment's context.  The experimental results describe the robot moving to the center of an H and an O, but the origin of these letters was not described.  The starting state of the experiments is also not clearly described (e.g. what was the robot's starting configuration in each episode).  The number of episodes used for training and evaluation were not described.  The network architecture was not described.  The contribution is not clear.

---

### Decision · Program_Chairs · 2018-03-20
**ICLR 2018 Workshop Acceptance Decision**

**Decision:**

Reject

**Comment:**

Based on the reviews, this paper has not been accepted for presentation at the ICLR workshop. However, the conversation and updates can continue to appear here on OpenReview.